# Photobiomodulation reduces hippocampal apoptotic cell death and produces a Raman spectroscopic "signature"

**David J. Davies**[1,2]*, **Mohammed Hadis**[3], **Valentina Di Pietro**[1], **Giuseppe Lazzarino**[4], **Mario Forcione**[1,2], **Georgia Harris**[5], **Andrew R. Stevens**[1,2], **Wai Cheong Soon**[1], **Pola Goldberg Oppenheimer**[5], **Michael Milward**[3], **Antonio Belli**[1,2], **William M. Palin**[3]

1 Department of Neuroscience and Ophthalmology, Institute of Inflammation and Ageing, University of Birmingham, Birmingham, United Kingdom, 2 National Institute of Health Research Surgical Reconstruction and Microbiology Research Centre, University Hospitals Birmingham' Edgbaston, Birmingham, United Kingdom, 3 Photobiology Research Group, School of Dentistry, College of Medical and Dental Science, Institute of Clinical Sciences, University of Birmingham, Birmingham, United Kingdom, 4 Department of Chemical Sciences, Laboratory of Biochemistry, University of Catania, Catania, Italy, 5 Faculty of Chemical and Biological Engineering, University of Birmingham, Birmingham, United Kingdom

* daviesdj@doctors.org.uk

**Data Availability Statement:** All relevant data are within the paper and its Supporting Information files.

## Abstract

Apoptotic cell death within the brain represents a significant contributing factor to impaired post-traumatic tissue function and poor clinical outcome after traumatic brain injury. After irradiation with light in the wavelength range of 600–1200 nm (photobiomodulation), previous investigations have reported a reduction in apoptosis in various tissues. This study investigates the effect of 660 nm photobiomodulation on organotypic slice cultured hippocampal tissue of rats, examining the effect on apoptotic cell loss. Tissue optical Raman spectroscopic changes were evaluated. A significantly higher proportion of apoptotic cells 62.8±12.2% vs 48.6±13.7% (P<0.0001) per region were observed in the control group compared with the photobiomodulation group. After photobiomodulation, Raman spectroscopic observations demonstrated 1440/1660 cm$^{-1}$ spectral shift. Photobiomodulation has the potential for therapeutic utility, reducing cell loss to apoptosis in injured neurological tissue, as demonstrated in this *in vitro* model. A clear Raman spectroscopic signal was observed after apparent optimal irradiation, potentially integrable into therapeutic light delivery apparatus for real-time dose metering.

## Introduction and background

Traumatic brain injury (TBI) is a significant contributor to global morbidity and mortality, affecting individuals of all ages [1]. After an initial traumatic injury to the brain, a significant number of neurological cells are lost to apoptosis contributing to a reduction in both neuronal and glial cell populations within the injured brain [2]. Apoptotic cells have been observed both in and around cerebral contusions (areas of necrotic cell death), as well as in brain regions remote to areas of structural pathology [3]. *In vitro* studies have identified that a threshold proportion of apoptotic cells in and around cerebral contusions (66.5% of the observed cellular

**Funding:** DD received a grant from the Midland Neuroscience Teaching and Research Fund (University of Birmingham Grant Number 1001177) (https://mntrf.org.uk/). The funders had no role in study design, data collection and analysis, decision to publish, or preparation of the manuscript.

**Competing interests:** The development of this novel concept has resulted in a patent pending application from our group relating to the invasive delivery of PBM, together with the use of temporarily implanted apparatus to establish an optimal dose feedback loop via an optical spectroscopic brain interface (UK Patent Application No 2006201.4). There are no other competing interests to declare, including those relating to employment, consultancy, other patents or products in development. The authors confirm that this does not alter our adherence to all PLOS ONE policies on sharing data and materials.

population) demonstrates a sensitivity of 89.5% and specificity of 66.7% in predicting mortality after severe TBI [3]. Cells undergoing apoptotic transformation (particularly those involved in ascending tracts of oligodendritic lineage [4]) have been observed in traumatic spinal cord injury, spreading to adjacent spinal segments [5], potentially contributing to the reduction in physiological function observed clinically.

Currently, there exist no clinically translated interventional treatments that are aimed at reducing the burden of neurological cell loss during the acute phases of TBI. The majority of clinical management strategies focus on the re-establishment of brain environmental homeostasis [6]: removing active compression, optimising perfusion and oxygenation [7], along with the restoration of functional anatomy. Any new therapeutic approach aimed at reducing the primary burden of cell loss (particularly due to apoptosis within the tissue injury penumbra [3]) has the potential to significantly improve the functional outcome of individuals following TBI.

Photobiomodulation (PBM) or 'low-level light therapy" (LLLT) is the therapeutic application of light for the purpose of facilitating healing and regeneration via any given light source, most frequently light-emitting diodes (LEDs) or a laser. Positive responses have been recorded with the use of light in the wavelength range of 600–1200 nm [8–10], with peak effects observed at approximately 660 nm.

Reduced wound healing time in superficial tissue, accompanied by less pain and inflammation, have been observed as beneficial effects of PBM [11]. Recent investigations have also been demonstrated to increase angiogenesis in wound repair and offer beneficial effects following myocardial infarction [12, 13]. PBM has also been widely used in dentistry due to established benefits of faster oral wound healing and pain relief [14, 15], and most notably with success in treating cancer patients with oral mucositis. Critically, PBM has demonstrated some positive effects in cerebral stroke models through inhibition of the secondary cascade and promoting neurogenesis [16].

A reduction in cell death due to apoptosis has been observed as an effect of PBM in a variety of models. A recent study reported attenuation of TNF/CHX induced apoptosis in squamous endothelial cells [17]. Similar reductions of programmed cell death were observed by a study into gingival fibroblast growth, where irradiated fibroblast colonies lost fewer cells to apoptosis compared with ambient light controls [18]. In vivo studies have also supported these findings, demonstrating attenuation of apoptosis in gastrocnemius myocytes following high intensity exercise in a murine model [19]. Other similar investigations have also observed a reduction of apoptosis in submandibular salivary glands in a diabetic rat model [20].

The exact mechanism by which PBM exhibits these positive biological responses is not fully understood, although it is proposed that the direct or indirect modulation of the enzyme cytochrome c oxidase (COX) is integral to the effect. COX is a large transmembrane protein complex located in the inner mitochondrial membrane [8]. Four metallic centres within its structure act as photoacceptors, transducing photosignals [8, 21]. COX stimulation using red and near-infrared (NIR) light causes the activity of the electron transport chain to increase, ultimately increasing the abundance of adenosine triphosphate (ATP) [8, 21, 22]. It is proposed that photons (of sufficient intensity and number) are absorbed by COX within the mitochondria [8, 22, 23]. This absorption potentiates the oxidative metabolic cascade resulting in beneficial effects of PBM [22, 24]. However, control cell lines and those lacking COX demonstrate cell proliferation [25], suggesting that some effects of PBM are via alternative pathways.

Initial studies investigating the effects of PBM on TBI within animal models have demonstrated potential translatable benefit including reduction in lesion size (up to 50%), improvements in objective neurological function, and reductions in observable trauma-related degeneration [26, 27]. Although not directly considered in these investigations, effects are

likely to be related to the reduction in apoptotic cell loss as demonstrated in non-TBI PBM studies [17, 18, 28].

The intensity (irradiance) and total time of light exposure have long been identified as key factors in the magnitude of the beneficial effect of PBM [29], however, there is little evidence in the literature to indicate the optimum dose for survival benefit in neurological tissue. The range of light intensities (irradiance ($mWcm^{-2}$), fluence ($J/cm^2$) and the dose (J) discussed in the generic literature leading to clinically significant effects on cell survival and tissue regeneration imply that transcranial delivery of the photons may not deliver a sufficient dose [30]. It may also be the case that the optimal photon dose varies between individual subjects, or different discrete regions of a given target tissue or organ.

Conceptually, the optical apparatus required for tissue photon delivery represent a potential for a simultaneous direct to tissue optical monitoring interface. Through simultaneous optical monitoring, the effect of PBM may be contemporaneously observed using a range of spectroscopic techniques, particularly Raman spectroscopy [31, 32]. Variations over time in the acquired optical spectra may represent a viable method of metering optimal dose and achieving the optimal desired effect.

Raman spectroscopy produces chemically specific optical signatures by utilising the inelastic scattering of coherent light, detecting the shift in wavelengths ($cm^{-1}$) observed following the energy change between light and matter [33]. Incident photons vibrate the molecules forming the target substance before scattering and the re-captured photons can be collected to form a spectrum of peaks indicating the biochemical structure of the substance [31, 32]. Raman spectroscopy for biological analysis typically use incident wavelengths in the visible/NIR region of the electromagnetic spectrum, with sensitivity to detected photons from biological molecules with wavenumbers in the region of 600–1800 $cm^{-1}$ and 2500–3400 $cm^{-1}$ [34]. Raman spectroscopy is sensitive, rapid and achieved without labelling or strict sample preparation [33]. Optimising the acquisition settings for biological samples would allow for non-destructive diagnoses or monitoring, applicable to intra-cranial access to brain tissue.

## Aims

1) To evaluate the potential of 660 nm PBM in reducing the quantity of brain tissue cells lost to apoptosis in a hippocampal organotypic slice culture model (as an *in vitro* TBI model). Here we will also investigate the influence of irradiance, the effect of irradiance and exposure time (total photon dose), and consecutive daily doses on the number of cells initiating programmed cell death within the cultured tissue.

2) To investigate changes in tissue optical (Raman) spectroscopic signature elicited by PBM along with its relation to photon dose, and from this consider if a specific signature in tissue Raman spectrum can be developed as a feedback marker to confirm optimal or adequate tissue dosage.

## Materials and methods

### 1) Sprague-Dawley rat hippocampus organotypic slice model

A hippocampectomy was performed immediately post-mortem on adult Sprague-Dawley rats under Home Office licence, sacrificed via carbon dioxide toxicity. Once performed, 150 μm hippocampal slices were separated (Fig 1). Individual slices were then placed onto polytetrafluoroethylene (PTFE) semi-permeable membranes (4 slices per membrane) incorporated into individual well inserts (Millicell® cell culture inserts, Millipore, PICM03050) fitting into a standard 6 well culture plate (Sigma®, SIAL0516). Glucose fortified B27-supplemented neurobasal medium (sNBA) solution (1 ml) was then added to each well. Plates containing the

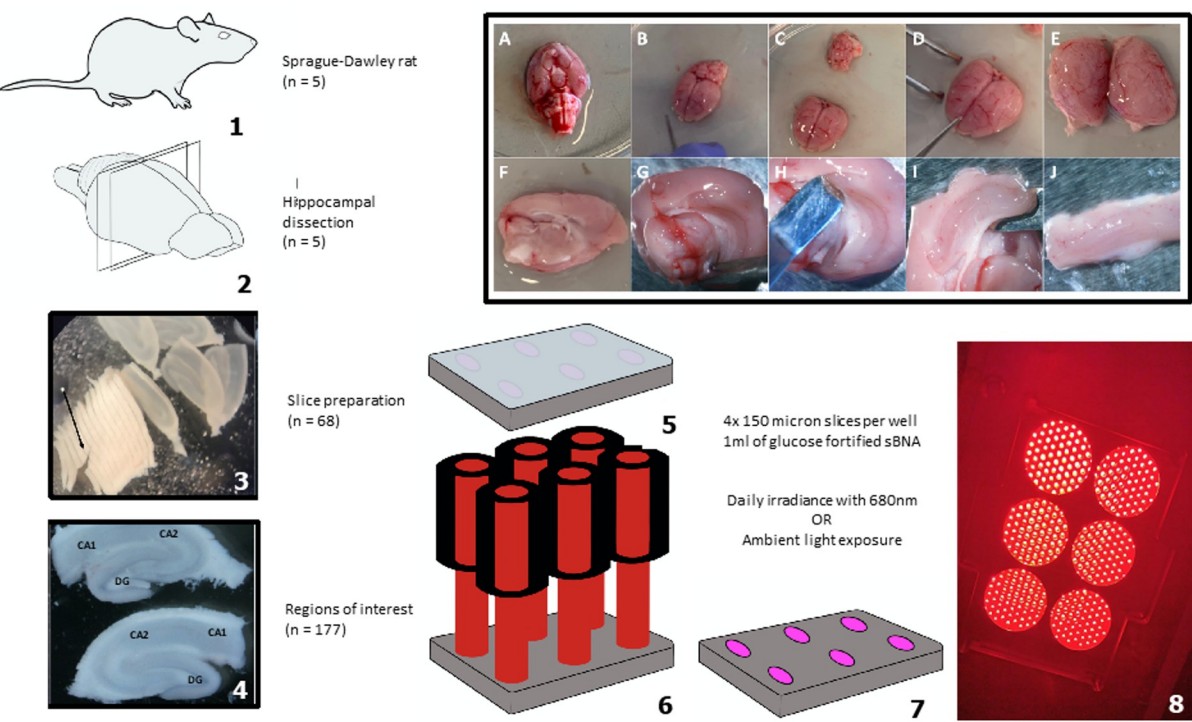

**Fig 1. Overview of organotypic hippocampal slice model preparation and administration of PBM.** (1) Sprague-Dawley rats were sacrificed; (2) hippocampal dissection (further in top right panel images A-J); (3) manual slice preparation; (4) demonstration of the regions of interest per slice (two slices shown); (5) standard six well culture plate; (6) culture plates exposed to irradiance; (7) culture plates exposed to ambient light; (8) photograph of LED arrangement for irradiance exposure.

slices were light protected using foil and incubated at 37˚C / 5% $CO_2$. The initial dissection and slicing of the tissue, particularly the manual slice separation serves as an in vitro analogue of TBI, not requiring the stable culture population and injury calibration of previously described 'stretch' models [35]. A daily visual assessment was undertaken. All subsequent analysis would be carried out on tissue and media harvested from this process, paired control and intervention samples were obtained from the same sacrificed specimens.

## 2) Photon dose delivery, beam profile and calibration

Therapeutic light of 660 nm wavelength was delivered in a single daily dose using a variable irradiance calibrated LED light source array (BioThor device, Thor Photobiomedicine). This wavelength was selected as it has demonstrated the greatest efficacy within the current limited subject literature [19–24]. Light intensity was controlled using a variable voltage power supply. Irradiance and beam homogeneity were confirmed via UV Vis Spectroscopy and beam profiling respectively.

**2.1. Spectrophotometric light characterisation.** The BioTHOR plate irradiator was spectroradiometrically characterised using a National Institute of Standards and Technology (NIST) calibrated fiber coupled spectrophotometer (USB400 UV-Vis Spectrometer, Ocean Optics) to obtain information on absolute irradiance and wavelength (Fig 2). Prior to calibration, the spectrometer was assembled with a 200 µM optical fiber and CC3 opal glass cosine corrector (3.9mm diameter). Following calibration, an empty 6-well plate was placed into the plate carrier with an aluminium 'mask' directly below the plate so only the area corresponding to the wells were exposed to light. The cosine corrector of the calibrated fiber was placed centrally into each

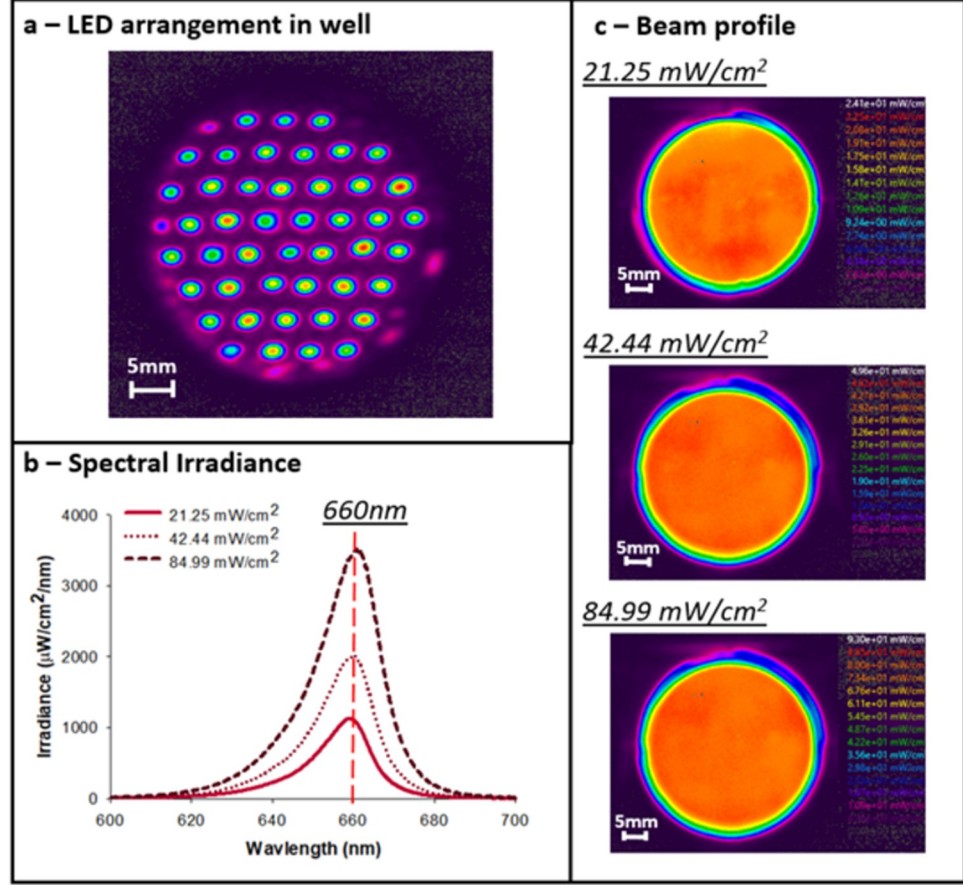

**Fig 2. Characterisation of the LED array to confirm spectral irradiance and beam homogeneity.** Each well is irradiated by 47 miniature LEDs with Gaussian beam profiles (a). However, at the surface of the 6-well plate a homogenous beam profile is produced due to divergence of the beams from the individual LEDs and scattering at the plate surface. This produces a uniform irradiance with an average irradiance of 21.25 mW/cm², 42.44 mW/cm² and 84.99 mW/cm² for the three tested light intensities (b and c) (measured at tissue level, with a beam diameter equal to well diameter = 34.8mm). Fluence values for 1 min exposure time: LT1 = 1.28 J/cm²; LT2 = 2.55 J/cm²; LT3 = 5.10 J/cm². Total power output of array (when delivering 42.44 mW/cm² at tissue level) = 4.244 W.

well of the 6- well plate so that the surface of the cosine corrector was in contact with the lower surface of the well to allow the measurement of the amount of light delivered during *in vitro* irradiation of hippocampal slices. Adjustment of the supply voltage to the array to pre-determined values supplied by the manufacturer allowed specific irradiances to be delivered to the plates. The spectral irradiance was recorded using Ocean View software (Ocean Optics, UK) and the absolute irradiance was calculated from the integral of the emission trace.

**2.2 Beam profile measurements.** The beam profile of the BioTHOR plate irradiator was determined using a CCD based beam profiler camera (Spiricon SP620, Ophir) following optical and linear calibration. The camera was focused onto the clear lower surface of a 6-well plate and beam profiles were measured either with or without a diffuser target screen (Opal glass target screen, Thorlabs) placed between the 6-well plate and light source. For each measurement, an ambient light correction was applied and images were recorded statically to assess the homogeneity of the beam delivered in each well.

**2.3 Dosing.** A commencing dose of 2 minutes irradiance at 42.4 mW/cm² daily was selected. The six well plate containing slices to be treated (intervention plates) was placed on

the source, covered over with an ambient shield. Control plates containing hippocampal slices from the same sacrificed animals (identically and contemporaneously prepared and incubated adjacent to the intervention plates) were placed in ambient light during treatment. All plates were foil shielded as soon as the light therapy was completed and then returned to the incubator.

### 3) ImmunoFluorescent cell imaging

NucView 488 Caspase-3 was utilised as a fluorescent apoptotic cell marker. Caspase-3 is a key enzyme within the apoptotic pathway, and once activated (as apoptosis is triggered) it cleaves the NucView 488 substrate, liberating the fluorescent product. On the final day of culture, the media within each well was replaced with 1 ml of media containing Nucview 488 (1/200 dilution e.g 5 μl NucView in 995μl glucose fortified sNBA). Plates were then incubated for a further 4 hours. The well media was then replaced with 1 ml 4% paraformaldehyde (PFA) and incubated for 20 minutes shielded from light at room temperature. The membrane inserts containing hippocampal slices then underwent three consecutive washes in phosphate buffered saline (5 minutes per wash). These were then removed from their respective wells and mounted using VECTASHIELD® antifade mounting medium containing 4′,6-diamidino-2-phenylindole (DAPI) for the purpose of additional discreet nuclear marking. A Zeiss axioscope was used for the purpose of fluorescent imaging. Three specific and consistently recognisable regions were imaged on each hippocampal slice, the dentate gyri, and the coru ammonis (CA) regions 1 + 2. Three images were obtained for each region without Z stack: an image showing the total number of cells (DAPI fluorescent blue channel); an image showing only the apoptotic cells (NucView 488 related green channel); and a merged image to observe the relative ratio of these. Images were discarded where image quality was deemed poor or corrupted with artefact or inclusion.

For live cell imaging experiments (investigating the effect of consecutive daily PBM doses), imaging took place daily. Media was replaced each day with 1 ml media containing the Nucview 488 and pure DAPI (5 μl NucView and 1 μl pure DAPI in 994 μl sNBA). Plates were then incubated at 37˚C, 5% $CO_2$ for 4 hours. Following incubation, images were taken as per the above protocol, however without removal of the membranes or fixation. The fluorescent media was then removed and replaced with sNBA. As a contrast agent for cell nuclei was not applied, only absolute numbers of cells entering (caspase-mediated) apoptosis are obtained. The next PBM dose was then applied where applicable. Plates were then returned to the incubator (37˚C, 5% $CO_2$).

Images obtained were then analysed and automatic cell counts acquired using ImageJ (NIH, University of Wisconsin, USA). For each slice in all three regions (DG, CA1, CA2), the total cell numbers (DAPI blue channel fluorescence) and apoptotic cells (NucView) were counted providing a percentage ratio of cells that had initiated apoptotic cell death.

### 4) Tissue Raman spectroscopy

Immediately after the final PBM or control treatment in each individual culture experiment, individual slices retrieved from the culture wells were placed onto an aluminium backing plate (for optical noise reduction) and gentle pressure exerted on the tissue sample directly to create a (visually) homogenous Raman scanning surface of uniform thickness. Spectroscopy was carried out using an inVia™ confocal Raman microscope with incorporated spectrometer (Renishaw, Wotton-under-Edge, UK), The integrated 'WiRE' software package (Renishaw, Wotton-under-Edge, UK) was used for acquisition and image processing. After surface focusing using 20x objective magnification, a 633 nm laser at 100% device specific power was used

to obtain spectra. A total of 3 x 6 second exposures were obtained to formulate the definitive spectrum from each hippocampal slice. Acquisition and processing were carried out in line with previous investigations undertaken [32, 36]. Based on previous investigation the peak intensity at 1440 cm$^{-1}$ and 1660 cm$^{-1}$ were the focus of analysis along with their respective ratios (due to expected variability in raw data absolute quantity).

## 5) Statistical analysis

Data was assessed for normal distribution (Shapiro-Wilk) and resultantly where data was non-parametric, a Mann-Whitney statistical test was used to ascertain the significance between the two considered continuous variables (intervention and control). A paired student's t-test was utilised for statistical analysis in the case of normally distributed data (effect of PBM on apoptosis in intervention and control samples), with matched intervention/control samples taken from a single sacrificed specimen. A $P$ value < 0.05 was considered statistically significant within this context. Tukey HSD was utilised for multiple comparisons (cumulative effect of daily exposure). Statistical analysis was performed using SPSS (IBM 2019).

## 6) Animal research

All experimental protocols were approved by the University of Birmingham and performed under the Home Office licence held by our department. All experiments were performed in accordance with relevant guidelines and regulations. Reporting of methods follows the recommendation of the ARRIVE guidelines.

## Results

### 1) Initial effect

An initial control group of 28 slices (82 observed regions of interest (ROI)) was compared with a light irradiated group of 40 slices (95 ROI) from sacrificed animals (n = 5). An initial 2 minutes of PBM at was applied daily for 5 consecutive days at an irradiance of 42.4 mW/cm$^2$ (LT2). On average, significantly more apoptotic cells 62.8±12.2% vs 48.6±13.7% ($P$<0.0001) per region were observed in the control group compared with the PBM irradiated group (Fig 3). The individual slice with the highest percentage of apoptotic cells (85.6%) occurred within the control group. Conversely, the irradiated group contained the slice with the lowest percentage of apoptotic cells (14.1%). A differential benefit (in terms of proportion of cells lost to apoptosis) was also observed between individual hippocampal regions (Fig 4).

### 2) Effect of irradiance

To assess the effect of irradiance, 5 day cultures with daily (2 minute) treatments at an irradiance of 21.3 mW/cm$^2$ (LT1), and 85.0 mW/cm$^2$ (LT3) were undertaken for comparison with the initial 42.4 mW/cm$^2$ (LT2). LT1 and LT3 were applied in a total of 20 and 24 hippocampal slices ($n$ = 57 and $n$ = 67 ROI, respectively) along with 18 contemporaneous control slices ($n$ = 35 ROI) from sacrificed specimens ($n$ = 4). Within the LT1 treated slices 45.8±12.7% of observable cells underwent apoptosis, significantly less than the 62.8±12.2% observed in the control specimens (Fig 5; $P$ < 0.0001). Within the LT2 treated slices the observable apoptotic cell population was 43.9±14.6% and 54.7±11.9% in the LT3 group, both significantly lower than control (Fig 5; $P$ = 0.0018 and $P$ < 0.0001, respectively).

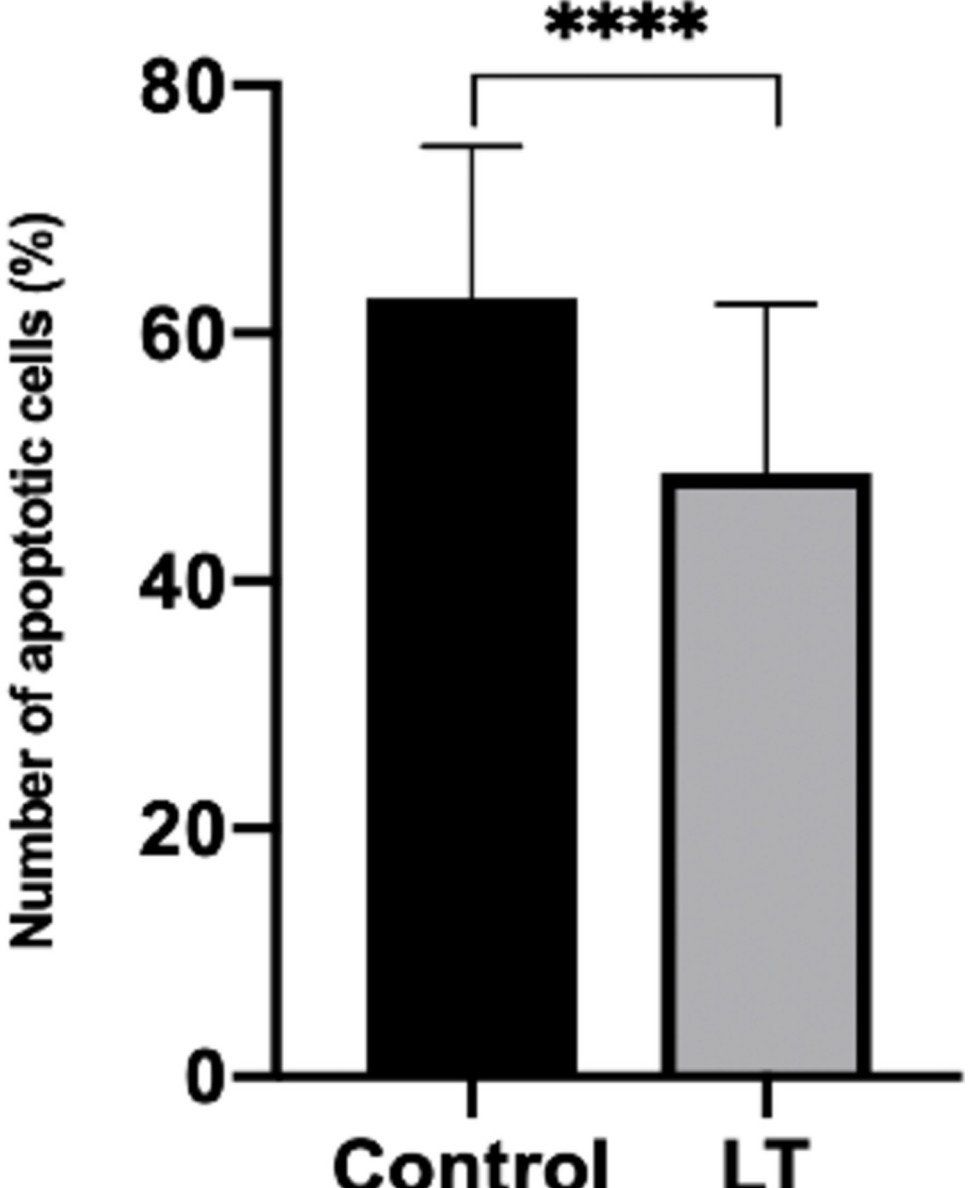

**Fig 3. Initial observed effect in the reduction of cell loss to apoptosis.** LT = light therapy (PBM). **** denotes
$P < 0.0001$, error bars represent standard deviation (SD).

### 3) Effect of dose duration

Daily doses of 1 minute, 2 minutes (initial) and 3 minutes were applied to the 5 day slice culture model. In total, 14 slices ($n = 41$ ROI) received 1 minute of irradiation a day, 14 slices received 2 minutes ($n = 39$ ROI) and 15 slices ($n = 44$ ROI) received 3 minutes. The previously ascertained optimal intensity of 42.44 mW/cm$^2$ was applied, with a contemporaneous control group of 15 slices ($n = 43$ ROI) cultured from a total of 3 sacrificed specimens ($n = 3$). The mean proportions of apoptotic cells observed in control slices was 60.2±11.0% compared with 51.6±17.0% in the 1 minute group ($P = 0.0066$), 54.6±13.5% in the 2 minute group ($P = 0.0437$), and 61.4±15.2% in the 3 minute group ($P = 0.182$) (Fig 6).

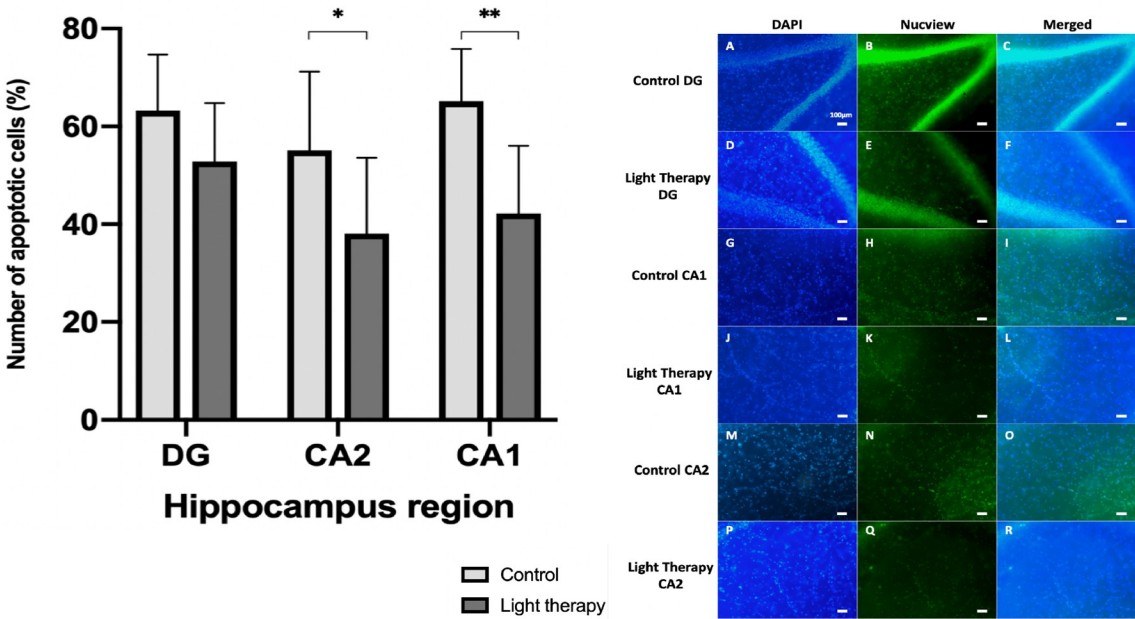

**Fig 4. Differential effect of LT2 on the individual hippocampal regions observed.** LT2 = 660nm irradiation for two minutes per day over a 5 day culture. ** denotes $P < 0.01$, * denotes $P < 0.05$, error bars represent SD. CA1 = cornu Ammonis 1, CA2 = cornu Ammonis 2, DG = dentate gyrus, DAPI = 4', 6-diamidino-2-phenylindole.

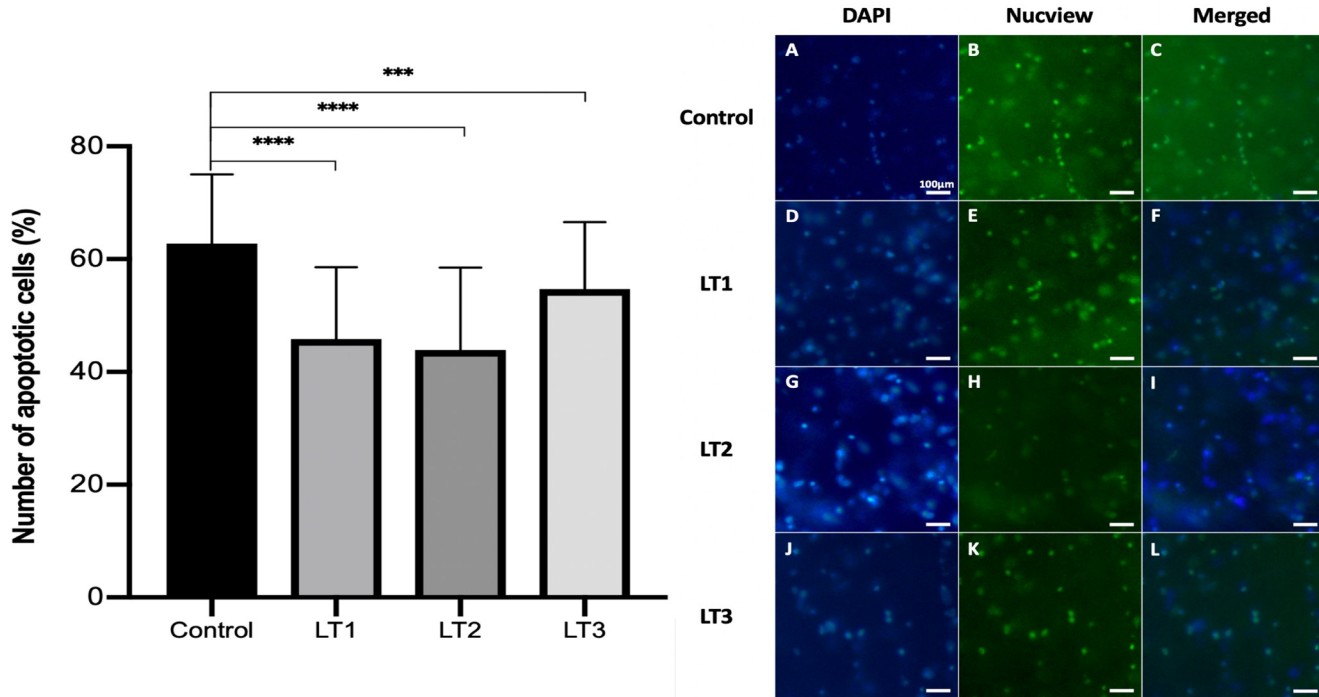

**Fig 5. Effect of irradiance—% of apoptotic cells in each group, LT1 = 21.3 mW/cm², LT2 = 42.4 mW/cm², LT3 = 85.0 mW/cm².** **** denotes $P < 0.0001$, *** denotes $P < 0.001$, ** denotes $P < 0.01$, error bars represent SD. DAPI = 4', 6-diamidino-2-phenylindole.

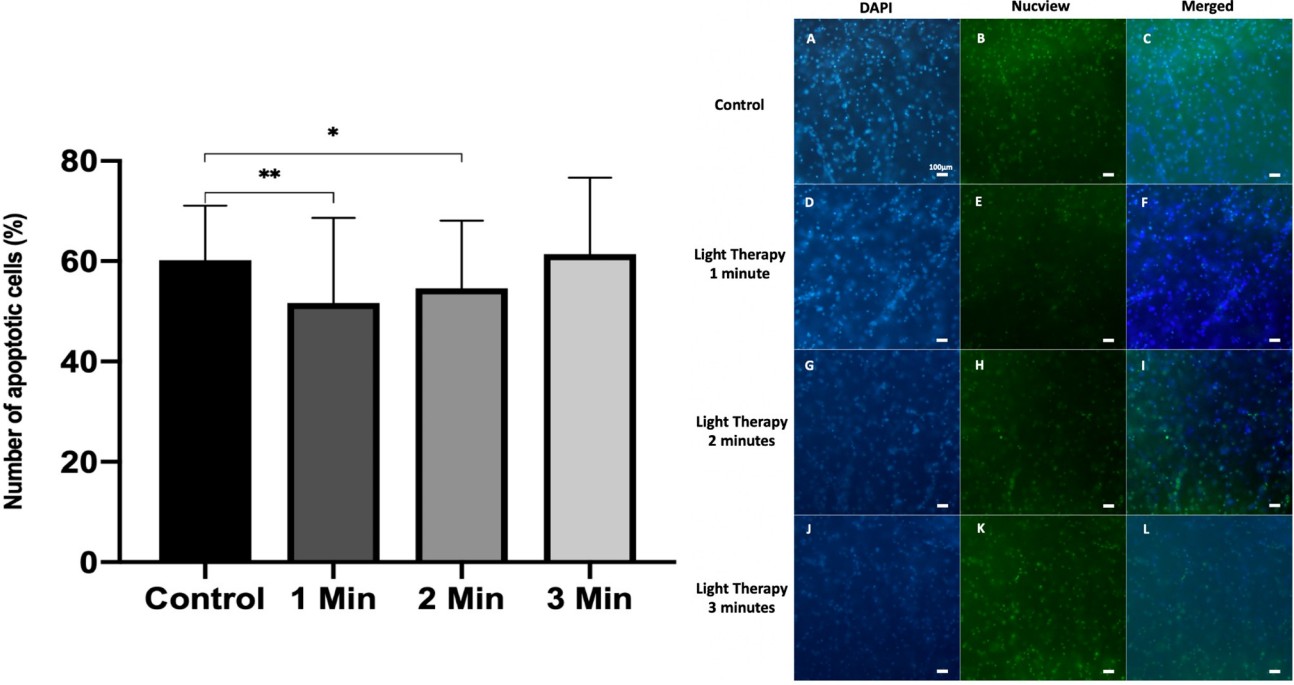

**Fig 6. Percentage of apoptotic cells in response to varying light therapy durations, either 1, 2 or 3 minutes per day over 5 days.** ** denotes P < 0.01, * denotes P < 0.05, error bars represent SD. DAPI = 4', 6-diamidino-2-phenylindole.

## 4) Cumulative effect of daily doses

A simultaneous 5 day culture with daily live imaging was undertaken with: (1) a control group of 10 slices (no light therapy, $n$ = 27 ROI), a 1 day group of 10 slices (receiving one dose of light therapy 0 h after injury and no further doses, $n$ = 27 ROI), a 2 day group of 10 slices (receiving two doses of light therapy at 0 h and 24 h after injury then no further doses, $n$ = 30 ROI), a 3 day group of 9 slices (receiving three doses of light therapy at 0 h, 24 h and 48 h after injury and no further doses, $n$ = 25) and a 4 day group of 10 slices (received four doses of light therapy at 0 h, 24 h, 48 h and 72 h after injury, $n$ = 28 ROI). Slices were taken from 4 sacrificed animals ($n$ = 4). Daily treatment given was 1 min exposure to LT2.

The control group of slices showed a steady reduction in apoptotic cell death over the 4 day experiment (Table 1). After 2 days, no group of cultured slices demonstrated a significant reduction in apoptotic cells compared with the control. At 3 and 4 days, intervention groups had significant apoptosis reduction compared with the control group ($P$ = 0.039 and $P$ = 0.008 respectively). A cumulative beneficial effect was demonstrable (Fig 7) on daily doses.

**Table 1. Proportion of apoptotic cell loss (%) at 24, 48, 72 and 96 hours in control (no PBM) 1, 2, 3, and 4 consecutive days of PBM.**

|  | 24 hrs | 48 hrs | 72 hrs | 96 hrs |
|---|---|---|---|---|
| **Control** | 67.4±6.10% | 64.7±4.02% | 63.6±5.60% | 62.4±4.00% |
| **1 day** | 62.6±15.2% | 62.1±5.05% | 61.3±3.39% | 60.4±3.84% |
| **2 day** | 64.2±5.20% | 59.2±8.59% | 60.2±3.09% | 59.7±3.84% |
| **3 day** | 62.2±6.84% | 58.6±5.54% | 56.7±6.05% | 56.8±3.83% |
| **4 day** | 64.7±7.73% | 61.8±4.55% | 57.1±5.80% | 54.7±5.27% |

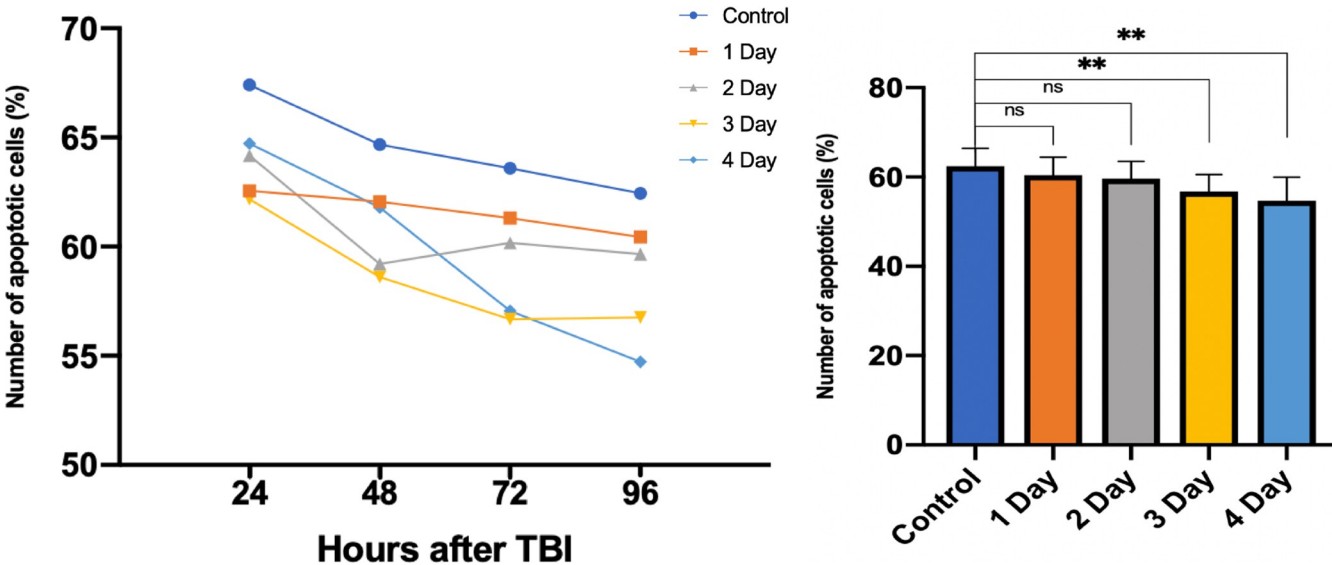

**Fig 7. Effect of daily doses of irradiation on cell loss of LT2 (42.4mW/cm$^2$) for 1 minute per day, for 1 to 4 days.** ns denotes no significance, ** denotes P < 0.01, error bars represent SD.

### 5) Raman spectroscopic signature of effect

After a 5 day organotypic culture, 36 slices underwent Raman spectroscopy (*n* = 18 control and *n* = 18 receiving LT2 irradiance therapy for 1 minute per day). For each of the spectra produced, the shift corresponding to the largest peaks in the control condition were 1440±0.7 cm$^{-1}$ and 1659±0.8 cm$^{-1}$. In the treatment condition, they were at 1440±0.6 cm$^{-1}$ and 1658±0.6 cm$^{-1}$. For brevity, the peaks in both groups were approximated to 1440 cm$^{-1}$ and 1660 cm$^{-1}$, respectively. In line with previously undertaken investigations [32, 36] focus centred chiefly on the magnitude of shift at these points in the acquired Raman spectra (Fig 8). Average intensity (peak size) at 1440 cm$^{-1}$ was 1407±187 (au) and 1288±110 in the control and PBM conditions, respectively. The 1660cm$^{-1}$ peak had a mean intensity of 1090±130 in the control samples, and 1291±66 in the treatment samples (Fig 6). There was a significant change in the average 1440/1660cm$^{-1}$ peak ratio between the therapy and control groups (Mann-Whitney U, *P* = 0.0204), with the ratio increasing from 0.774 in the control group to 1.002 in the therapy group (24.8% relative increase in ratio).

An additional cohort of 14 slices was prepared. 7 of these underwent a 3 day slice culture with daily PBM at LT1 irradiance for 1 minute prior to Raman analysis, with the remaining 7 slices progressing to a 5 day (daily 1 minute LT1) culture before spectroscopic examination in order to observe the progression of the ratio over multiple daily doses. This culture was undertaken with an identical number of matched control slices. Here, an increase in 1440cm/1660 cm$^{-1}$ ratio from the 3 day culture samples vs controls (0.757 vs 0.891; 17.7% increase, *P* = 0.092) was observed, along with a further increase in the ratio after 5 days. No significant difference was observed between the 1440cm/1660cm$^{-1}$ peak ratios at each therapeutic interval (3 day vs 5 day; *P* = 0.4).

### Discussion

The framework of this series of experiments is based on exploring the possibility of PBM as a clinically viable tool to improve cell survival within the context of moderate to severe TBI, with real-time Raman spectroscopic dose metering. The delivery of such a clinical tool would be

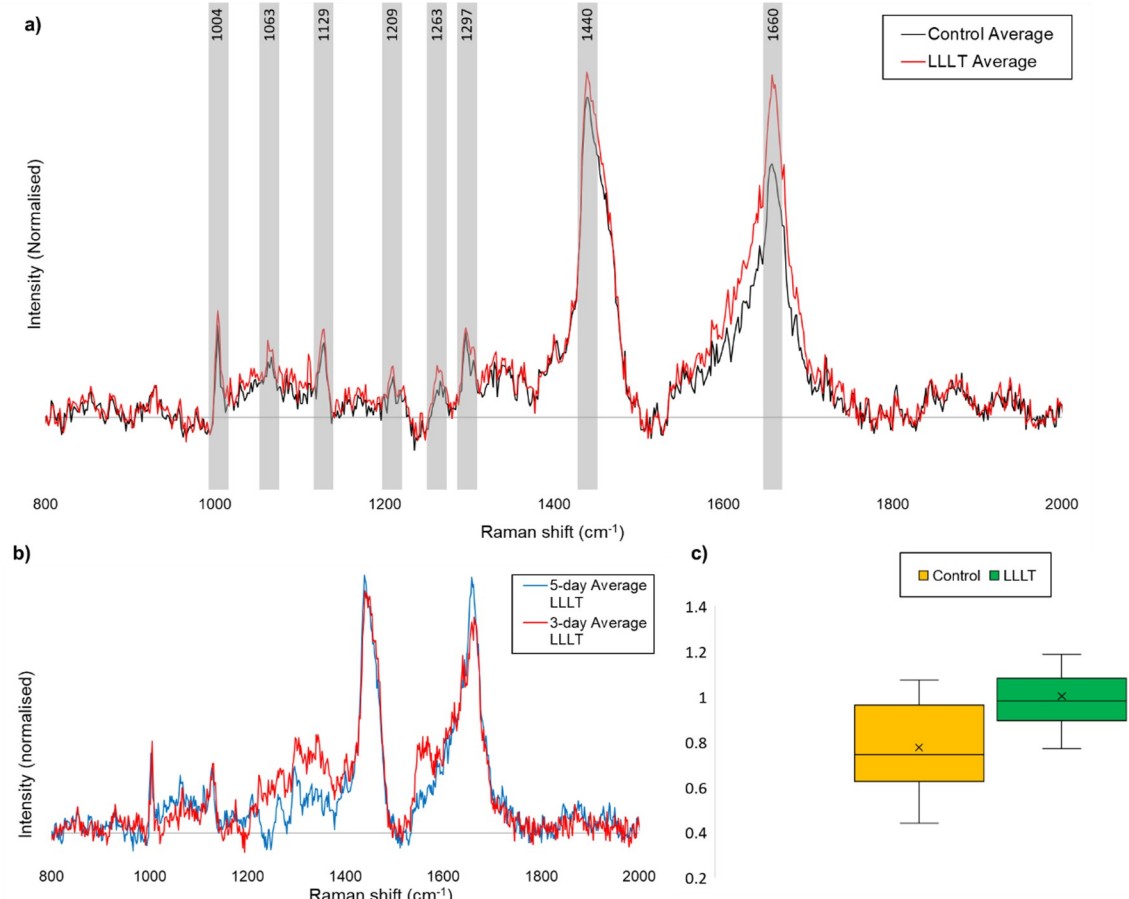

**Fig 8. a)** Comparative figure of average control and average PBM Raman spectra normalised, illustrating the changes in average intensity of the 1440 and 1660 cm$^{-1}$ peaks, and highlighting the assigned, characteristic peaks of rat hippocampus. **b)** Comparative figure of normalised Raman spectra from average 3-day PBM samples and average 5-day PBM samples, no significant difference was observed between the 1660 cm$^{-1}$/1440 cm$^{-1}$ peak ratios at each therapeutic interval (p = 0.4). **c)** Box and Whisker plot illustrating the increase in average peak intensity ratio of 1660 cm$^{-1}$/1440 cm$^{-1}$, increasing from 0.774 for control samples to 1.002 for PBM samples (p = 0.0204).

heavily determined by the irradiance and duration of light exposure required to produce a clinically meaningful effect.

Irradiance with 660 nm light has demonstrated a significant improvement in cell survival. An exposure of 1 minute at an irradiance of 42 mW/cm$^2$ had the greatest decrease in the number of apoptotic cells (an absolute reduction of 18.9%). This work has determined a biphasic pattern of irradiation dose: an increase from approximately 42 mW/cm$^2$ led to an apparent reduction in effect. However, halving the dose demonstrated a statistically similar effect. Although the range of tested intensities and durations was limited and definitive quantities impossible to deduce, it may be reasonable to consider the optimal irradiance as between 21 and 42 mW/cm$^2$. However, more investigation would be required to establish this precisely.

Similarly, when assessing the effect of the duration of exposure, 1 minute demonstrated the greatest magnitude of reduction in programmed cell death. Extension of duration to 2 and 3 minutes diminished the observed positive effects. From the observations on intensity and duration, we can infer that it is likely that a certain rate of delivery and/or a total number of absorbed photons will exert the greatest positive effect, although this investigation did not address the effects of exposure reciprocity, i.e. the potential to observe similar benefits for a

prolonged duration at a reduced intensity, or an increased irradiance for a shorter time. These additional required investigations would provide evidence as to whether the total number of photons absorbed was the true proportional variable to effect magnitude.

Live daily image assessment demonstrated a positive cumulative effect of daily application of 1 minute PBM at the optimally established irradiance of approximately 42 mW/cm$^2$. Daily application led to a progressive reduction in programmed cell death. The reduction in apoptosis within the live daily imaged control samples (reduction in live cell stock within the organotypic slice culture translating to a lower absolute number) was further decreased by the application of daily doses of PBM. When daily applications ceased, the trajectory of reduction returned to that of the control (Fig 5), suggesting that any observed positive effect is transient.

A further observation can be made from the differential benefit between hippocampal regions; the greatest magnitude of effect seemed to be realised by regions of lower mitotic activity and cell turnover [37]. This may have implications on PBM having the greatest benefits in cell preservation in tissues that have naturally low levels of mitosis/cellular division (the cerebral cortex).

Application of red/NIR PBM to neuronal cell culture has a background in the literature, and the findings here correlate with previous study. In SH-SY5Y cells, 635 nm light (18 J) has been shown to reduce mitochondria dependent apoptosis after sodium nitroprusside exposure [38]. Giuliani et al. have previously demonstrated that 670 nm at 0.45 mJ/cm$^2$ stimulate neurite outgrowth and increased cell viability in conditions of oxidative stress in rat pheochromocytoma cells (PC12) [39]. In primary astrocyte cell culture, 660 nm light at 6 J/cm$^2$ was found to promote cell proliferation [40]. Using 670 nm light, with Sommer and colleagues demonstrated that PBM significantly increased cell proliferation in amyloid β stressed human neuroblastoma cells [41].

A fundamental limitation of this investigation is the organotypic model utilised. Here we have no quantified injury burden (as would be in previously described 'stretch' models [35]), and rely on the process of preparation together with the naturally suboptimal conditions of the in vitro environment to model the tissue injury. As a result, within the following hours and days, a sustained reduction in the number of cells is expected. The trajectory and 'gradient' of this rate of cell death will be steepend or flattened depending on any additional noxious or nourishing modifications to the culture environment. A quantified and reproducible insult, requiring a prolonged stable (without progressive and terminal loss of surviving cell stock) slice culture [42] was not reliably achievable within our laboratory infrastructure. Unacceptable levels of contamination and sample loss was experienced, leading to inconsistency and smaller data samples.

The Raman spectra recovered and compared between both control and intervention slices revealed a clear signature of tissue response to PBM (as tested at 42 mW/cm$^2$ intensity for 1 minute per day). All characteristic peaks (identified and assigned in Table 2) are larger in magnitude for the average spectra of PBM samples than control, representing a greater number of protein, lipid, amino acid and glycogen molecular bonds, indicating fewer apoptotic cells. Due to Raman intensity having arbitrary units, and the expected variability in raw data absolute quantity, peak intensity between separate samples are not an ideal source of comparison. The most consistently comparable feature between samples in these spectra is the ratio between the largest peaks: at 1660 cm$^{-1}$ and 1440 cm$^{-1}$.

The 1660 cm$^{-1}$ peak (Amide I region) is proposed to correspond with the number of protein bonds in the tissue. An increase in this peak intensity (increased presence of protein bonds) as seen in PBM spectra, corresponds with decreasing numbers of apoptotic cells. Conversely, the 1440 cm$^{-1}$ peak is proposed to represent cholesterol bonds, which shows only a modest increase in intensity in PBM compared to control in our samples. Relative stability of the 1440

**Table 2. Raman peak assignment for characteristic peaks identified in rat hippocampi spectra [43–47].**

| Peak Wavenumber (cm⁻¹) | Assignment | Origin |
|---|---|---|
| 1004 | $\nu$(C-C) ring | Phenylalanine |
| 1063 | $\nu$(C-C) lipids | Phospholipids, Aliphatic side chains |
| 1129 | $\nu_s$(COC), $\nu_s$(PO₂), $\nu$(CN), $\nu$(CC) | Glycogen, DNA, Phenylalanine, Lipids, Aliphatic side chains |
| 1209 | $\nu$Ph, δCHC, CH₂ wagging and twisting | Hydroxyproline, Tyrosine, Tryptophan, Phenylalanine |
| 1263 | Amide III, CH₂ twisting | Lipids |
| 1297 | δ(CH₂) lipids, Amide III | Aliphatic side chains, Lipids |
| 1440 | δ(CH₂) lipids and proteins, CH₂ twisting and bending | Cholesterol, Phospholipids, Tyrosine, Proteins |
| 1660 | $\nu$(C = C) lipids, C = C stretching Amide I | Tyrosine, Lipids, Proteins, Alpha-helix/random coil |

$\nu$ = stretching; $\nu_s$ = symmetric stretching; δ = in-plane bending, Ph = Phenyl.

cm⁻¹ peak between PBM and control permits its use as a reference point for the 1660cm⁻¹ intensity, hence the 1440cm/1660cm⁻¹ ratio.

Corresponding survival data was not acquired in this experiment: future investigation incorporating both data streams focusing on potential correlation between the magnitude change/ratio change in these peaks and the proportion of cells initiating apoptotic (and potentially necrotic) cell death is required. The implications of these observations are two-fold: firstly, it would provide a potential avenue for real time monitoring of the effects of PBM within the in vivo and/or clinical setting. Observing a shift in peak ratio may provide insight into when an optimal photon dose of PBM has been delivered, removing the risk of over- or under-dosing which would lead to a reduction in potential positive effect. Secondly, this shift in 1440/1660cm⁻¹ ratio may have the potential to provide an independent measurement of brain injury burden and prognostic potential, allowing a real time roadmap of tissue recovery together with information regarding the effectiveness of other therapeutic interventions. Similar approaches have been undertaken previously to quantify the redox state of components in the mitochondrial cytochrome electron transport chain [48]. However, additional work into the optical Raman spectra of tissue irradiated by PBM is required to provide insight into the fundamental effect of light on cellular metabolism, giving a definitive answer to the mechanisms underpinning this photo-cellular effect.

A considerable strength of the present work amongst the literature is the rigorous accuracy in radiometry and dosing [49, 50]. This has been achievable through a collaborative approach between research groups, combining expertise in both photobiomodulation and neural repair. Further cross-disciplinary working has facilitated the integration of optical monitoring techniques into the model, which has demonstrated a spectroscopic "signature" for monitoring dose-effect. This modality for real-time feedback of effect alongside accurate radiometry has allowed development of a system which has the potential to permit therapeutic titration of PBM in a manner which has not been described previously.

Whilst previous work has adopted a transcranial approach to light delivery [16], irradiance within this order, and Raman spectroscopic monitoring, would require a direct-to-brain surface or intra-parenchymal approach at one or more points in order to achieve a meaningful effect. Together with the proposed dosing schedule, and should these observations be confirmed in further in vivo investigation, intracranial access of such a device would be required for sufficient delivery. The integration of an appropriate optical interface into a device delivering the current standard of care is potentially viable, posing a minimal risk burden to the treated individual. The current standard of care for the management of these conditions

involves direct intracranial access, and the placement of an array of catheters into the brain parenchyma substance [51, 52]. Access to this is achieved via burr holes in the skull and is undertaken in all cases meeting the minimum severity requirement [52] with or without the requirement for surgical intervention. Intracranial light delivery in *in vivo* Parkinsonian models has been shown successful previously [53–55], demonstrating feasibility.

The development of this novel concept has resulted in a patent pending application from our group relating to the invasive delivery of PBM, together with the use of temporarily implanted apparatus to establish an optimal dose feedback loop via an optical spectroscopic brain interface (UK Patent Application No 2006201.4). The conceivable optical apparatus required to deliver a meaningful dose of PBM directly onto or into the brain substance (fibre optical), could be utilised for the Raman spectroscopic examination of the brain surface (an emission and detection fibre) and integrated into a single device [56]. As PBM treatment is only potentially required for a short time (1 minute) each day, the scope for establishing a spectroscopic interface to monitoring pathology, metering PBM dose and also tracking the deposition of pharmacological therapeutic agents is an exciting potential prospect for TBI care.

## Conclusion

The application of 42 mW/cm$^2$ of 660 nm light for 1 minute a day has a significant effect on reducing the number of cells lost to apoptosis in the organotypic slice culture considered here. The effects are cumulative daily. A clear Raman spectroscopic signature is observable and may provide a reliable biofeedback mechanism for the metering of optimal dose. The technology to integrate potential direct-to-brain PBM delivery together with real time biofeedback should be the subject of future investigation together with further research into its mechanistic underpinnings.

## Acknowledgments

We thank James Carroll, THOR Photomedicine for supplying the BioTHOR device used in this study.

## Author Contributions

**Conceptualization:** David J. Davies, Andrew R. Stevens, Pola Goldberg Oppenheimer, Michael Milward, Antonio Belli, William M. Palin.

**Data curation:** David J. Davies, Mohammed Hadis, Valentina Di Pietro, Giuseppe Lazzarino, Mario Forcione, Georgia Harris, Andrew R. Stevens.

**Formal analysis:** David J. Davies, Valentina Di Pietro, Giuseppe Lazzarino, Mario Forcione.

**Funding acquisition:** David J. Davies.

**Investigation:** David J. Davies, Mohammed Hadis, Valentina Di Pietro, Giuseppe Lazzarino, Mario Forcione, Georgia Harris.

**Methodology:** David J. Davies.

**Project administration:** David J. Davies, Mohammed Hadis.

**Resources:** Mohammed Hadis, Michael Milward, William M. Palin.

**Software:** Mohammed Hadis, William M. Palin.

**Supervision:** Pola Goldberg Oppenheimer, Michael Milward, Antonio Belli, William M. Palin.

**Validation:** Mohammed Hadis.

**Writing – original draft:** David J. Davies, Andrew R. Stevens, Wai Cheong Soon.

**Writing – review & editing:** David J. Davies, Mohammed Hadis, Andrew R. Stevens, Wai Cheong Soon, Michael Milward, William M. Palin.

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
