## [Decision Letter · Decision Letter 0]

2 Feb 2022

PONE-D-22-01807Photobiomodulation reduces hippocampal apoptotic cell death and produces a Raman spectroscopic “signature”PLOS ONE

Dear Dr. Stevens,

Thank you for submitting your manuscript to PLOS ONE. After careful consideration, we feel that it has merit but does not fully meet PLOS ONE’s publication criteria as it currently stands. Therefore, we invite you to submit a revised version of the manuscript that addresses the points raised during the review process.

Minor points raised by reviewer 2.

We look forward to receiving your revised manuscript.

Kind regards,

Michael R Hamblin

Academic Editor

PLOS ONE

Journal Requirements:

2. We note that you have a patent relating to material pertinent to this article. Please provide an amended statement of Competing Interests to declare this patent (with details including name and number), along with any other relevant declarations relating to employment, consultancy, patents, products in development or modified products etc. Please confirm that this does not alter your adherence to all PLOS ONE policies on sharing data and materials, as detailed online in our guide for authors http://journals.plos.org/plosone/s/competing-interests by including the following statement: "This does not alter our adherence to  PLOS ONE policies on sharing data and materials.” If there are restrictions on sharing of data and/or materials, please state these. Please note that we cannot proceed with consideration of your article until this information has been declared.

Reviewers' comments:

Reviewer's Responses to Questions

**Comments to the Author**

1. Is the manuscript technically sound, and do the data support the conclusions?

Reviewer #1: Yes

Reviewer #2: Yes

2. Has the statistical analysis been performed appropriately and rigorously? 

Reviewer #1: Yes

Reviewer #2: Yes

3. Have the authors made all data underlying the findings in their manuscript fully available?

Reviewer #1: Yes

Reviewer #2: Yes

4. Is the manuscript presented in an intelligible fashion and written in standard English?

Reviewer #1: Yes

Reviewer #2: No

5. Review Comments to the Author

Reviewer #1: This in vitro study of photobiomodulation therapy (PBMT) for reduction of hippocampal cell apoptosis is of good quality. I have no complaints. I think this preclinical paper contributes to the field of PBMT research.

Reviewer #2: The current manuscript is an interesting in vitro study regarding neuroprotective effects of red PBM on neuronal cell culture. Along with observed anti-apoptotic effects, a Raman spectroscopic signature was obviously observable. This probably can provide a reliable biofeedback mechanism for the PBM real-time dosimetry.

Here are some comments:

On line 30, for the sentence, you can cite recently published papers (e.g., PMID: 29164625; PMID: 29327206).

The English writing quality of the manuscript should be improved.

On line 76, PBM parameters such as irradiance, fluence, the dose should be reported as follow, respectively, mW/cm2, J/cm2, and J. The use of "irradiance, fluence, the dose" should be consistent throughout the manuscript.

On line 104, Irradiance refers to "Power density" not "Photon density".

On the paragraph starting from line 89, the functional wavelength spectrum of the Raman spectroscopy could be mentioned. In other words, this technique is sensitive to what particular range of the spectrum (e.g, visible, near-infrared, mid-infrared, far-infrared, etc)?

On line 153, "irradiance (power)" should be changed to "irradiance (power density)".

PBM parameters should be completely reported; Output power, fluence, beam diameter and area, beam profile (Gaussian or Hot top), the distance of the LED probe from the dishes, etc.

In the Discussion section, some of the previous in vitro studies on positive neuroprotective effects of 2.5 J/cm2 red light on neuronal cell culture can be discussed (see PMID: 33935090).

6. PLOS authors have the option to publish the peer review history of their article (what does this mean?). If published, this will include your full peer review and any attached files.

Reviewer #1: No

Reviewer #2: **Yes: **Farzad Salehpour

---

## [Author Response · Author response to Decision Letter 0]

9 Feb 2022

Dear Prof. Hamblin,

On behalf of all of our co-authors, I would like to extend our sincere thanks to you and the reviewers for the time and consideration of our article. We were greatly encouraged to read the positive responses from the reviewers on the scientific merit of our submitted work.

We have taken time to consider and act upon the suggestions for improvement offered to us by Reviewer 2, and we enclose an updated manuscript as requested which reflect the changes. The authors agree that the updated manuscript, with the changes described below, represents a considerable improvement to the original work, and we offer our further thanks for such comments for improvement.

The response to each suggestion is enclosed below in bold type face, with comments above in italics, and numbered for ease of reference. 

1. On line 30, for the sentence, you can cite recently published papers (e.g., PMID: 29164625; PMID: 29327206).

Response: These citations have been added

2. The English writing quality of the manuscript should be improved.

Response: The document has been reviewed in its entirety, with minor changes made to grammar, phrasing and syntax which we hope the reviewers will agree has made a considerable improvement to the writing quality, without affecting the scientific content. 

3. On line 76, PBM parameters such as irradiance, fluence, the dose should be reported as follow, respectively, mW/cm2, J/cm2, and J. The use of "irradiance, fluence, the dose" should be consistent throughout the manuscript. 

Response: This sentence has been updated, and this nomenclature is reflected where mentioned and throughout the manuscript.

4. On line 104, Irradiance refers to "Power density" not "Photon density".

Response: We are in agreement with the reviewer that irradiance is not referring to “photon density”. The term “irradiance” has now been left, without further qualification, given clear establishment of terms relating to PBM parameters as suggested in point 3.

5. On the paragraph starting from line 89, the functional wavelength spectrum of the Raman spectroscopy could be mentioned. In other words, this technique is sensitive to what particular range of the spectrum (e.g, visible, near-infrared, mid-infrared, far-infrared, etc)?

Response: We have added a further sentence which clarifies the typical wavelengths used in Raman spectroscopy for biological samples, and the respective wavenumbers which are detected and typically represent spectra from biological molecules. We feel this is an adequate though succinct summary which addresses the point without over-elaboration on RS to a non-specialist readership. 

6. On line 153, "irradiance (power)" should be changed to "irradiance (power density)".

Response: As discussed in the response to point 4, we have changed this to: “irradiance”

7. PBM parameters should be completely reported; Output power, fluence, beam diameter and area, beam profile (Gaussian or Hot top), the distance of the LED probe from the dishes, etc.

Response: We have added further detail as suggested. Irradiance values given have been measured at the level of the tissue plane, and this has been made clear in the revised manuscript and is now accompanied by respective fluence values for 1 min treatment time. With the images in figure 2, and with the well diameter, beam diameter and fluence values now given, we feel this is a complete reporting of the salient parameters (as outlined in our group’s previous work on the topic (PMID: 29694800).

8. In the Discussion section, some of the previous in vitro studies on positive neuroprotective effects of 2.5 J/cm2 red light on neuronal cell culture can be discussed (see PMID: 33935090).

Response: We thank the reviewer for their suggestion and have included a short discussion with reference to these articles.

Yours sincerely

Mr David Davies

---

## [Editor Report · Decision Letter 1]

14 Feb 2022

Photobiomodulation reduces hippocampal apoptotic cell death and produces a Raman spectroscopic “signature”

PONE-D-22-01807R1

Dear Dr. Stevens,

We’re pleased to inform you that your manuscript has been judged scientifically suitable for publication and will be formally accepted for publication once it meets all outstanding technical requirements.

Kind regards,

Michael R Hamblin

Academic Editor

PLOS ONE
---

## [Editor Report · Acceptance letter]

18 Feb 2022

PONE-D-22-01807R1 

Photobiomodulation reduces hippocampal apoptotic cell death and produces a Raman spectroscopic “signature” 

Dear Dr. Stevens:

I'm pleased to inform you that your manuscript has been deemed suitable for publication in PLOS ONE. Congratulations! Your manuscript is now with our production department. 

Kind regards, 

on behalf of

Dr. Michael R Hamblin 

Academic Editor

PLOS ONE